# RELATIVE-BASED SCALING LAW FOR NEURAL LANGUAGE MODELS

## ABSTRACT

Scaling laws aim to accurately predict model performance across different scales. Existing scaling-law studies almost exclusively rely on cross-entropy as the evaluation metric. However, cross-entropy provides only a partial view of performance: it measures the absolute probability assigned to the correct token, but ignores the relative ordering between correct and incorrect tokens. Yet, relative ordering is crucial for language models, such as in greedy-sampling scenario. To address this limitation, we investigate scaling from the perspective of relative ordering. We first propose the Relative-Based Probability (RBP) metric, which quantifies the probability that the correct token is ranked among the top predictions. Building on this metric, we establish the Relative-Based Scaling Law, which characterizes how RBP improves with increasing model size. Through extensive experiments on four datasets and four model families spanning five orders of magnitude, we demonstrate the robustness and accuracy of this law. Finally, we illustrate the broad application of this law with two examples, namely providing a deeper explanation of emergence phenomena and facilitating finding fundamental theories of scaling laws. In summary, the Relative-Based Scaling Law complements the cross-entropy perspective and contributes to a more complete understanding of scaling large language models. Thus, it offers valuable insights for both practical development and theoretical exploration.

## 1 INTRODUCTION

Scaling laws are an important tool in the era of large language models. Their primary goal is to predict how model performance changes as the model size increases. (Hestness et al., 2017; Kaplan et al., 2020; Rosenfeld et al., 2020; Henighan et al., 2020) The key challenge of scaling law studies is identifying reliable performance metrics that can be accurately predicted. (Hoffmann et al., 2022; Bergsma et al., 2025) To date, cross-entropy has proven to be the most reliable metric for this purpose, and consequently it has become the dominant choice in scaling law research. Cross-entropy-based scaling laws not only guide the training of large language models, but also provide insights about model mechanisms and artificial intelligence theories. (Kaplan et al., 2020; Hoffmann et al., 2022; Henighan et al., 2020) Cross-entropy has even been adopted beyond language modeling to search for scaling laws in new domains, such as multimodal learning and information retrieval. (Shukor et al., 2025; Edwards et al., 2024; Aghajanyan et al., 2023; Lourie et al., 2025; Fang et al., 2024; Shukor et al., 2025)

However, focusing solely on cross-entropy as the metric provides an incomplete picture of a model's scaling behavior. This limitation arises because cross-entropy primarily measures the absolute probability assigned to the correct answer, while ignoring the relative ordering of predictions. (Chung et al., 2022; Xu et al., 2024a;b) In fact, absolute-based and relative-based perspectives capture two distinct aspects of model performance, and neither can substitute for the other. As illustrated in Figure 1, a model assigns a probability of 0.28 to the correct token. Yet the rank of this token can be different. In one case, two incorrect candidates may still outrank the correct one; in another, all incorrect candidates may receive lower scores, placing the correct token at the top. Thus, the same probability score may correspond to different rankings. Consequently, cross-entropy–based scaling laws fail to capture how the relative position of the correct answer changes with model size. This shortcoming is particularly severe because relative ordering plays a central role in practical applications of language models, such as greedy decoding and top-k sampling. (Noarov et al., 2025;

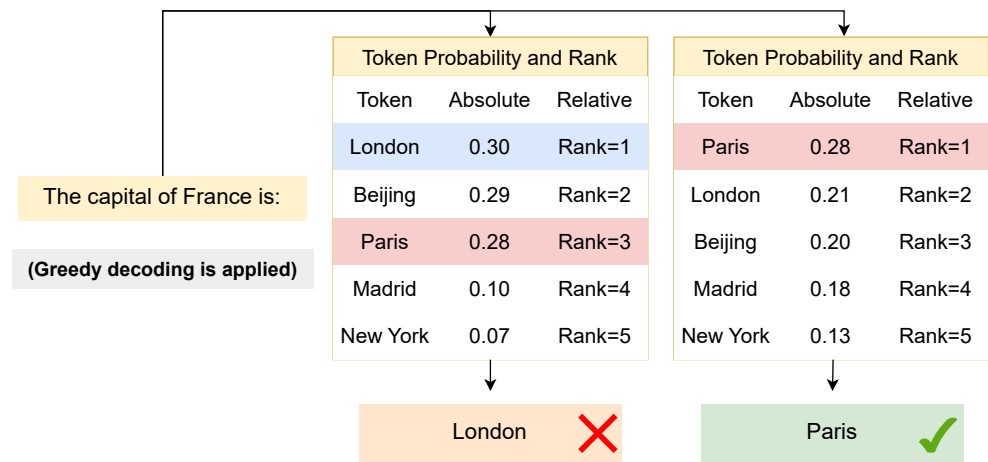

Figure 1: Illustration of the limitation of absolute-based metrics in evaluating generation performance. **(Left)** The ground-truth token has a higher absolute probability but is ranked below competitors, leading greedy decoding to fail. **(Right)** With a lower absolute probability yet a higher rank, the ground-truth token is correctly chosen. This shows that absolute-based metrics cannot capture the crucial relative ranking among tokens.

Lai et al., 2024; Bruch, 2019; Tang et al., 2024) The inability to account for such order-sensitive sampling strategies highlights a significant gap between cross-entropy metric and the performance observed in real-world usage.

To address this limitation[1], we introduce a new metric, Relative-Based Probability (RBP), to capture a model's ability to rank the correct token among the top candidates. Given a parameter $k$, $\text{RBP}_k$ measures the probability that the correct token appears within model's top-$k$ predictions. The computation of $\text{RBP}_k$ proceeds as follows: First, we select a corpus. Then, for each target token in the corpus, we check whether it is included among the top-$k$ tokens predicted by the model. Finally, we calculate the proportion of such cases over the entire dataset, which yields the RBP value. Unlike cross-entropy that is concerned with the absolute prediction value, RBP focuses on the relative ordering of predictions. As a result, $\text{RBP}_k$ provides a complementary perspective on model performance.

Next, we examine how $\text{RBP}_k$ changes as the model size increases. Our results reveal a clear scaling law when $k$ is much smaller than the vocabulary size. Specifically, we find that

$$-\log\left(\text{RBP}_k\right) \propto M^{-\alpha}, \quad k \ll \text{Vocab Size}, \tag{1}$$

where $M$ denotes the number of model parameters and $\alpha$ is a positive constant. We refer to this relationship as the Relative-Based Scaling Law. To investigate this relationship, we conduct experiments across four datasets. We use 4 model families, covering a total of 23 models whose sizes span over 5 orders of magnitude. The results show that when $k$ lies in the range of roughly 1 to 100, the relationship between $-\log(\text{RBP}_k)$ and model size follows a precise power-law trend. The fitted power-law coefficients reach approximately 0.99, which is on par with the coefficients observed in cross-entropy–based scaling laws. Relative-Based Scaling Law deepens our understanding of scaling behavior: enlarging model size fundamentally reshapes the ranking of tokens. As models grow, a greater proportion of correct tokens are placed among the top predictions, and this improvement follows a power-law trend. Such an understanding goes beyond what cross-entropy–based scaling laws alone can capture.

We believe Relative-Based Scaling Law holds important implications for the development of large language models as well as for advancing the theoretical foundations of artificial intelligence. In this work, we illustrate its utility through two concrete examples. (1) Relative-Based Scaling Law

---

[1]Code is available at `https://anonymous.4open.science/r/relative-based-scaling-law-7F3E`.

enhances our understanding of emergence phenomena: Prior studies have attempted to explain emergence through cross-entropy–based scaling laws, but such explanations fail to generalize to decoding strategies like greedy sampling or top-$k$ sampling, where relative ordering is crucial. (Wei et al., 2022; McKenzie et al., 2023; Schaeffer et al., 2023; Lu et al., 2023; Krakauer et al., 2025) By contrast, the Relative-Based Scaling Law naturally accounts for these scenarios and resolves this issue. (2) Relative-Based Scaling Law opens new directions for theoretical research: Although RBP and cross-entropy capture complementary aspects of model performance, we find that their scaling behaviors are surprisingly similar. They share not only identical mathematical form but also very close fitted exponents. This confusing coincidence suggests the existence of a more fundamental theory of intelligence that can unify both scaling laws. At present, no established theory can fully explain this phenomenon. In this paper, we put forward a conjecture as a step toward such unification.

In summary, Relative-Based Scaling Law provides a relative-ordering perspective on scaling behavior. It complements the cross-entropy view and contributes to a more complete picture of scaling in large language models. Together, the two laws shall offer both practical guidance for scaling up language models and theoretical insight into the fundamental theories of artificial intelligence.

## 2   RELATED WORK

Scaling laws for neural language models, often based on absolute metrics like cross-entropy loss ($\mathcal{L}_{\text{CE}}$), provide a framework for understanding performance evolution with model size, dataset size, and compute budget Kaplan et al. (2020); Henighan et al. (2020); Hoffmann et al. (2022); He et al. (2024); Edwards et al. (2024); Aghajanyan et al. (2023); Lourie et al. (2025). Notably, the "Chinchilla" laws emphasize balancing model and dataset size for compute efficiency (Hoffmann et al., 2022).

Absolute-based metrics, however, do not capture the relative ranking of the ground-truth token among candidates, which can misalign metric improvements with generative performance (Tang et al., 2024; Chatzi et al., 2024; Freitag & Al-Onaizan, 2017; Acosta et al., 2024; Song et al., 2024; Prabhu, 2024). To address this, relative-based metrics and differentiable top-$k$ losses have been proposed, improving rank-sensitive accuracy in both small- and large-scale models (Petersen et al., 2022; Xiao et al.).

Despite these advances, relative-based evaluation has not been systematically studied under scaling laws for generative language models. Our work fills this gap by introducing a relative-based metric and empirically analyzing its scaling behavior across architectures and datasets.

## 3   RELATIVE-BASED METRIC

In this section, we introduce our metric for evaluating model performance. Prior work has commonly relied on an *absolute-based* approach, which measures the probability assigned to the ground-truth token. In contrast, we propose a new *relative-based* metric, $\text{RBP}_k$, which evaluates whether the ground-truth token appears within the model's top-$k$ predictions. This provides a perspective that is complementary to cross-entropy and is more directly aligned with practical decoding strategies.

### 3.1   PRIOR ABSOLUTE-BASED METRIC

In previous studies, model performance has typically been evaluated using cross-entropy loss ($\mathcal{L}_{\text{CE}}$), which exhibits clear scaling behavior across model sizes. Let $t$ be the ground-truth token, and $\text{p}(t)$ be the probability score output by the model. Cross-entropy equals

$$\mathcal{L}_{\text{CE}} = \mathbb{E}[-\log \text{p}(t)], \tag{2}$$

However, $\text{p}(t)$ alone does not reflect the *relative ordering* of the ground-truth token among all candidate tokens. As shown in Figure 1, the ground-truth token may be assigned relatively high probability but not ranked as the top-1 candidate. If greedy-sampling or top-k sampling are used during inference, such a probability score cannot reflect models' real-world performance.

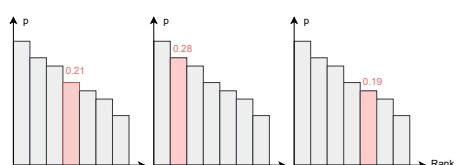
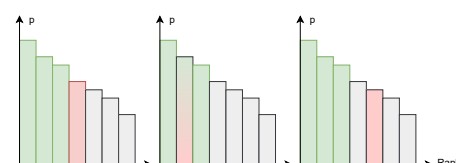
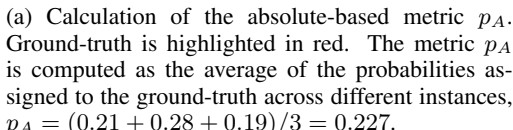

(a) Calculation of the absolute-based metric $p_A$. Ground-truth is highlighted in red. The metric $p_A$ is computed as the average of the probabilities assigned to the ground-truth across different instances, $p_A = (0.21 + 0.28 + 0.19)/3 = 0.227$.

(b) Calculation of the relative-based metric $\text{RBP}_k$ with $k = 3$. Ground truth is highlighted in red. The red-to-green gradient shows when the ground-truth falls within the top-$k$ ranked predictions. Since the ground-truth falls within the top-3 for only one of the three instances, $\text{RBP}_3 = 1/3 \approx 0.333$.

Figure 2: Illustration of two distinct metric calculation methods. Figure 2a shows the absolute-based metric $p_A$, which directly quantifies the probability assigned to the ground-truth. Figure 2b shows the relative-based metric $\text{RBP}_k$, which measures how often the ground-truth token falls within the top-$k$ predictions. These figures highlight how the two metrics capture different aspects of model behavior.

## 3.2 OUR RELATIVE-BASED METRIC

To address this problem, we introduce a *relative-based* metric, namely Relative-based Probability. It is computed based on a hyper-parameter $k$. Therefore, we denote this metric as $\text{RBP}_k$. $\text{RBP}_k$ measures the probability that the ground-truth token appears within the model's top-$k$ predictions. Compared to cross-entropy, it is an independent metric that complements the relative-ordering perspective to model performance. For example, if $\text{RBP}_1$ is 30%, it means that greedy sampling works for 30% cases. Yet this cannot be deduced from the cross-entropy value.

$\text{RBP}_k$ is formally defined as follows. Let $\mathcal{V}$ denote the vocabulary and $t$ be the ground-truth token. We use $p(v)$ be the score of token $v$. Then, the rank of ground-truth token $R$ equals:

$$R = \sum_{v \in \mathcal{V}} \mathbf{1}\{p(v) \geq p(t)\}. \tag{3}$$

$R$ is a random variable. We define $\text{RBP}_k$ as its cumulative distribution function. Formally, it equals:

$$\text{RBP}_k = \Pr(R \leq k), \tag{4}$$

We can see that $\text{RBP}_k$ is the probability that the ground-truth token is within the top-$k$ scored tokens.

In practice, we compute $\text{RBP}_k$ as follows. First, we select a corpus. Then, for each token in the corpus, we record $1$ if the ground-truth token falls in the top-$k$ predictions and $0$ otherwise. The average of these values yields an empirical estimation of $\text{RBP}_k$.

In the next section, we will show that $\text{RBP}_k$ also exhibits a power-law scaling behavior with model size when $k$ is small. Therefore it is able to precisely capture performance improvements from a relative-ordering perspective.

## 4 RELATIVE-BASED SCALING LAW

Based on Relative-based Probability, this section establishes the *Relative-based Scaling Law*. It characterizes how $\text{RBP}_k$ evolves with model size. Our finding is that for small $k$ values ($k \ll |\mathcal{V}|$, with $|\mathcal{V}|$ denoting the vocabulary size), $\text{RBP}_k$ follows a precise power-law relationship with model size:

$$-\log \text{RBP}_k \ \propto \ S^{-\alpha} \quad \left(k \ll |\mathcal{V}|\right) \tag{5}$$

where $S$ denotes the model size and $\alpha > 0$ is the scaling exponent. We refer to this relationship as the *relative-based scaling law*.

Now we start to investigate the relationship between $\text{RBP}_k$ and model size. Since $\text{RBP}_k$ depends on the choice of threshold $k$, experiments are conducted in the following three regimes:

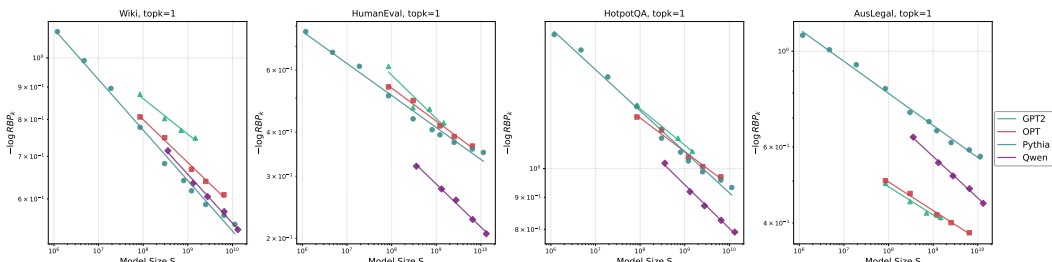

Figure 3: Relative-based scaling laws when $k = 1$. Across all model series and all datasets, $\text{RBP}_1$ exhibits precise power-law relationship with model sizes.

1. $k = 1$ regime: This regime is corresponding to the hardest setting where only the top-ranked token is considered. In this case, $\text{RBP}_k$ closely links to greedy decoding strategy.

2. Moderate-$k$ regime: In this regime, $k$ is larger than 1 but far smaller than the vocabulary size. In our experiments, we set $1 < k \leq 100$. This setup closely links to top-k sampling strategies.

3. Large-$k$ regime: In this regime, $k$ approaches to the vocabulary size. In our experiments, we set $k$ to 20,000 or 30,000.

These three regimes provide a comprehensive view of how $\text{RBP}_k$ scales with model size. We present the experimental results respectively in the following three subsections.

### 4.1 $k = 1$ REGIME

In the case of $k = 1$, the relative-based metric exhibits a remarkably precise power-law trend across all model families and datasets. Figure 5 illustrates $\text{RBP}_1$ versus model size on four datasets. For data points in the same model family, they form a straight line on the log-log plot. This indicates that $\text{RBP}_1$ precisely follows a power-law relationship with model size.

The result marks the discovery of a novel scaling law. It means that scaling up models increases the probability that the correct token is ranked highest. And the increasing rate follows a power-law rate. This perspective cannot be deduced from prior cross-entropy scaling law. It demonstrates that scaling up models fundamentally changes the relative-ordering and results in more correct tokens ranked at the top. Since $\text{RBP}_1$ closely links to the greedy decoding strategies, it reflects that greedy decoding performance will improve if model is scaled up.

### 4.2 MODERATE-$k$ REGIME: $1 < k \leq 100$

Moreover, we also observe that the power-law relationship still holds for small $k$ values. We set $k$ to 10, 50, and 100. The results are illustrated in Figure 5. We can see that for all cases, the data points lie on a straight line in the log-log plot. The fitted curves achieve $R^2 \geq 0.97$ across all datasets, indicating that the power-law relationship remains highly robust in this regime.

The result means that Relative-based Scaling Law can generalize to small $k$ values. When model is scaled up, more and more correct tokens are ranked within the top-$k$ predictions. And the rate of improvement follows a power-law manner. Since $\text{RBP}_k$ measures the probability that the ground-truth token appears within the top-$k$ candidates, it closely links to top-$k$ decoding strategy. It indicates that top-k sampling result will be improved when model is scaled up, which is another perspective that cannot be deduced from cross-entropy scaling law alone.

### 4.3 LARGE-$k$ REGIME: $k \to |\mathcal{V}|$

As $k$ approaches the vocabulary size $|\mathcal{V}|$, the scaling behavior of $\text{RBP}_k$ breaks down. We set $k$ to $20,000$ and $30,000$, and illustrate the results in Figure 5[2]. We can see that the data points scatter

---

[2]For Pythia, OPT, and GPT-2 models, vocabulary sizes are 50,256, 50,272, and 50,257, respectively, such that the chosen $k$ values nearly cover the entire vocabulary; for Qwen models, vocabulary sizes are 151,936

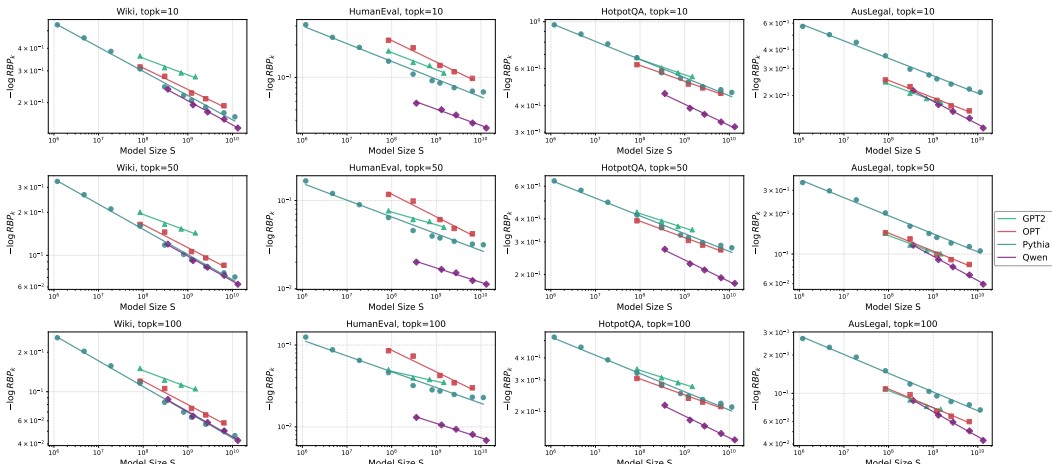

Figure 4: Scaling laws when $1 < k \ll |\mathcal{V}|$. The relative-based metric $\text{RBP}_k$ maintains strong power-law scaling behavior, with consistently high $R^2$ values across datasets and model series. This indicates that the metric reliably captures performance improvements under top-$k$ sampling strategies commonly used in practice.

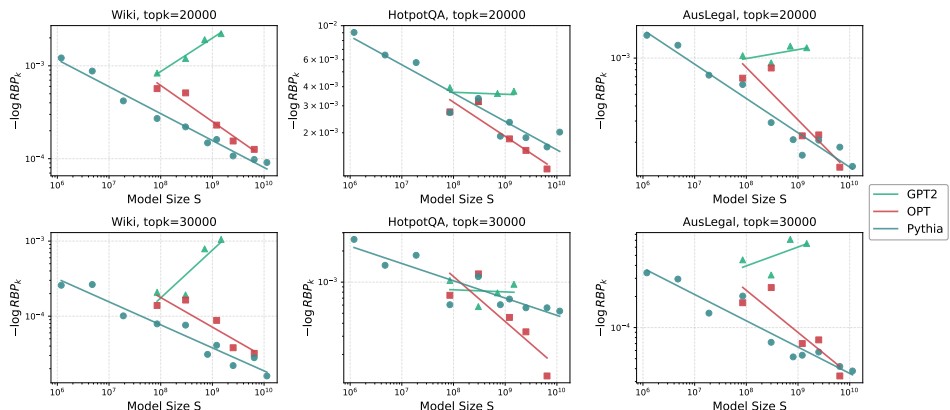

Figure 5: Scaling laws when $k \to |\mathcal{V}|$. In this regime, the power-law behavior deteriorates substantially, with large scatter and inconsistent slopes across models and datasets. This indicates that $\text{RBP}_k$ becomes less informative about scaling when the threshold approaches the vocabulary size.

significantly. For example, $\text{RBP}_k$ for GPT2 even increases rather than decreases. The case for $k = 30,000$ becomes much more random than the case for $k = 20,000$. Therefore, we believe that Relative-based Scaling Law only holds for $k$ that is small, typically under 100.

The reason for this behaviour remains to be investigated We conjecture that that reason might lies in random noise. When $k$ nearly covers the entire vocabulary, most ground-truth tokens are included in the top-$k$ set, regardless of model size. In this case, some noisy data may dominate $\text{RBP}_k$, which breaks down its power-law scaling pattern. The noise even makes $\text{RBP}_k$ increase when model is scaled up, as shown by the GPT2 results in Figure 5.

## 4.4 SUMMARY

The results across the three regimes collectively reveal how the choice of threshold $k$ governs the extent to which $\text{RBP}_k$ reflects power-law scaling with model size. In Figure 6, we summarize how the threshold $k$ affects the fitting quality and the scaling exponent.

---

for models up to 7B and 152,064 for models above 7B. We don't report Qwen series in this regime because it doesn't apply to the condition $k \to |\mathcal{V}|$.

According to the left figure, we can see that the scaling law holds robustly in the small and moderate $k$ regimes, where $k \ll |\mathcal{V}|$. The fitted quality only starts to drop after $k$ grows close to the vocabulary size. We can see that $R^2$ values remain high (above 0.9) when $k < 1000$. Therefore, Relative-based Scaling Law is a robust law for a wide range of $k$ values.

According to the right figure, the scaling exponent increases as $k$ increases. A higher exponent means that the performance increases faster when the model is scaled up. Therefore, the result indicates that optimizing $\text{RBP}_{100}$ shall be easy, since the exponent is close to $0.2$. Yet it is hard to optimize $\text{RBP}_1$, as the exponent is less than $0.1$. Such an understanding is a unique contribution of Relative-based Scaling Law.

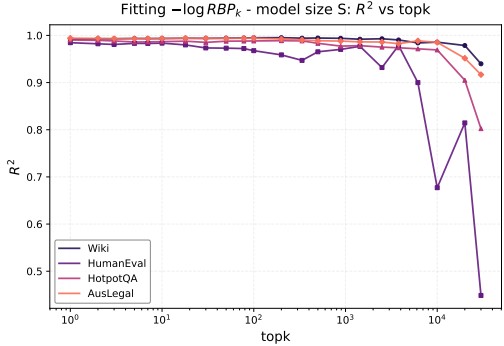 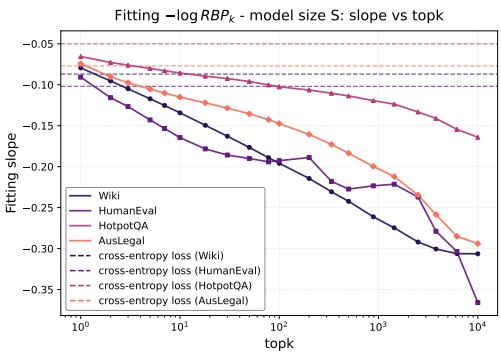

(a) Coefficient of determination ($R^2$) as a function of $k$. The scaling law holds robustly for $k < 1000$, where $R^2$ is very close to $1$. It starts to drop when $k$ is too large.

(b) Exponent of the fitted power-law as a function of $k$. Exponent for $k = 1$ is close to those of cross-entropy-based scaling laws. The The exponent value decreases gradually as $k$ increases, indicating that $\text{RBP}_k$ is easier to optimize for bigger $k$.

Figure 6: Scaling fitting results of $\text{RBP}_k$ as a function of $k$. Figure 6a shows the power-law fitting quality. Figure 6b shows how scaling exponent evolves with $k$.

## 5 APPLICATION OF RELATIVE-BASED SCALING LAW

In this section, we demonstrate two applications of Relative-based Scaling Law. The first is to explain emergence phenomena, and the other is to explore fundamental principles that drive both prior cross-entropy-based and our Relative-based Scaling Laws.

### 5.1 EXPLAINING EMERGENCE

A significant challenge for scaling laws is to explain complex macroscopic phenomena, such as the "emergence" of new capabilities in large language models. Emergence refers to the sharp, non-linear jump in performance on a specific task once model size surpasses a threshold. Although (Schaeffer et al., 2023) explain this phenomenon with cross-entropy-based scaling law, this scaling law cannot generalize to greedy or top-$k$ sampling and thus cannot explain emergence phenomenon in these scenarios. We show that the relative-based scaling law can complement this drawback and provide a quantitative explanation.

We define task success as the model correctly predicting $N$ consecutive ground-truth tokens within its top-$k$ candidates. Assuming independence and stationarity across positions, the success probability is:

$$p_{N,k} = (RBP_k)^N. \tag{6}$$

When $RBP_k \propto S^{-\alpha}$, the sequence-level relation becomes:

$$-\log p_{N,k} = -N \log(RBP_k) \propto N \cdot S^{-\alpha}. \tag{7}$$

Thus, sequence-level log-error scales smoothly with model size $S$, linearly amplified by $N$. Figure 7 confirms this prediction empirically.

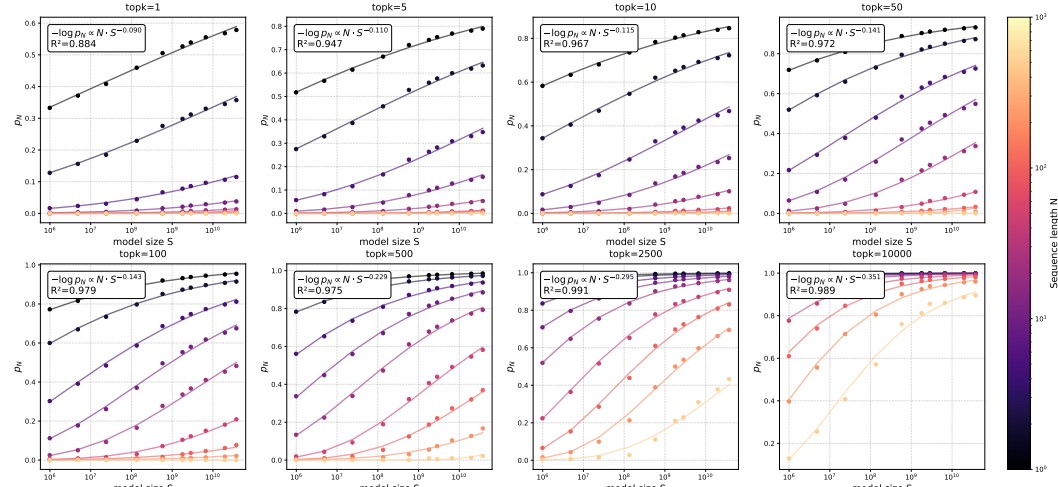

Figure 7: Empirical validation of the sequence-level scaling law, $-\log p_{N,k} \propto N \cdot S^{-\alpha}$. Each subplot corresponds to a fixed $k$, with colored lines for different $N$. The fitted power-law curves align closely with empirical data, confirming the law across a wide range of conditions. Crucially, the resulting curves naturally exhibit the sigmoidal shape characteristic of observed emergence phenomena, demonstrating that emergence follows directly from smooth microscopic scaling.

This explains the apparent "emergence." Since

$$p_{N,k} \approx \exp(-C \cdot N \cdot S^{-\alpha}), \tag{8}$$

for constant $C$, the probability curve is inherently sigmoidal:

- For small $S$, the exponent is large and negative, driving $p_{N,k}$ close to zero.
- As $S$ grows, the exponent approaches zero, and $p_{N,k}$ rises sharply toward one.

The sequence length $N$ acts as an amplifier: larger $N$ sharpens the transition, creating the observed sudden "knee" in performance.

In summary, emergence is not a breakdown of scaling laws but a predictable macroscopic effect of a smooth power law at the token level. The perceived phase transition is simply the exponential amplification of microscopic trends when mapped to sequence-level tasks under greedy or top-$k$ sampling.

## 5.2 CONNECTING CROSS-ENTROPY- AND RELATIVE-BASED SCALING LAWS

The next application is the peculiar connection between the Relative-based scaling law and cross-entropy-based scaling law. As emphasized earlier, the two laws capture fundamentally different aspects of the output distribution: cross-entropy-based law focuses on the probability mass assigned to the ground-truth token, while the relative-based law examines its rank among candidates. By construction, they are not interchangeable. Nevertheless, we observe that both exhibit remarkably similar power-law decay with respect to model size. In particular, when $k = 1$, the decay exponents of cross-entropy (absolute-based) and $-\log p_1$ (relative-based) are nearly identical, as shown in Figure 8a.

We believe this peculiar coincidence suggests a deeper principles that can derive both laws. Yet this perspective has not been proposed in prior theoretical studies and none of the existing theories can explain this. Therefore, we believe this coincidence shall be the key to the discovery of a deeper scaling theory.

In this paper, we propose a conjecture that can derive both scaling laws. We analyze the distribution of ground-truth token ranks across models of different sizes. Empirical evidence shows that these rank distributions are long-tailed. (Ma et al., 2025; Chatzi et al., 2024; Cai et al., 2024; Zhan et al.,

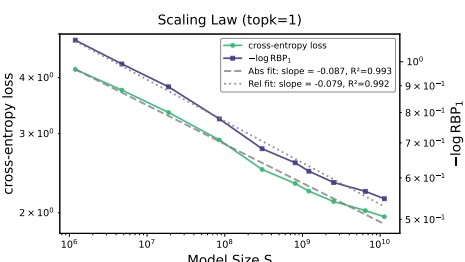

(a) Empirical results: Cross-entropy (CE) and $-\log(\text{RBP})_k$ exhibit similar scaling forms and slopes.

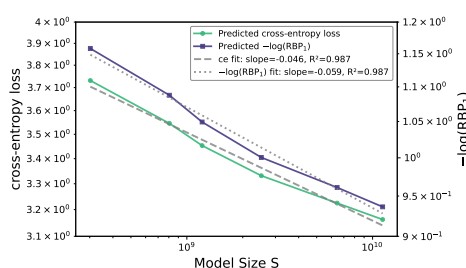

(b) Simulation results: CE and $-\log\text{RBP}_k$ show similar scaling patterns.

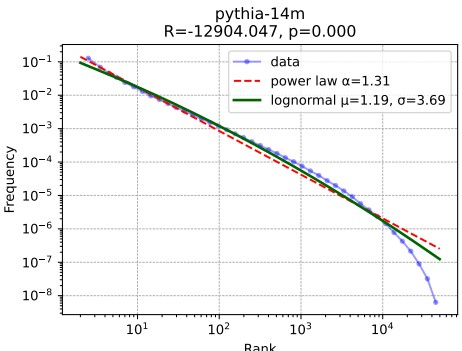

(c) Hypothesis 1: Ground-truth ranking frequency follows the lognormal distribution.

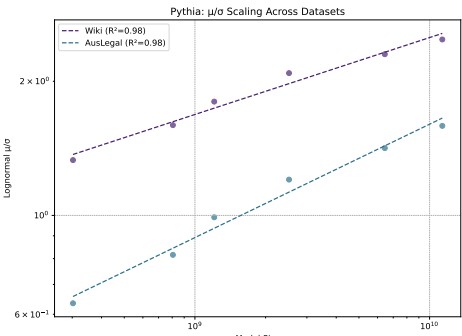

(d) Hypothesis 2: lognormal parameter $\mu/\sigma$ scales with model size.

Figure 8: Scaling behavior of cross-entropy (CE) and ranking-based losses.

2025). Therefore, our first hypothesis is that the rank distribution follows a lognormal distribution:

$$P(x) = \frac{1}{x\sigma\sqrt{2\pi}} \exp\left(-\frac{(\ln x - \mu)^2}{2\sigma^2}\right), \quad x > 0, \tag{9}$$

where $x$ denotes the rank of the ground-truth token, and $\mu$ and $\sigma$ are parameters that systematically depend on model size $S$. Figure 8c shows this is a good fit. Our second hypothesis is that when model is scaled up, the lognorm form stays the same while $\mu/\sigma$ scales with model size. This has also been observed in prior studies (Zhan et al., 2025) and empirically in Figure 8. Under the two assumptions, we can derive the two scaling laws, as shown in Figure 8b, which is similar to the real results, as shown in Figure 8a. This conjecture aims to show that the two laws together may indicate a deeper theory. For future scaling law theories, they shall not only aim to derive cross-entropy-based scaling law but also Relative-based Scaling Law with the same scaling exponent.

## 6 CONCLUSION

In this paper, we propose Relative-based Probability (RBP) and establish the Relative-Based Scaling Law to study model performance from the perspective of relative ordering. Unlike cross-entropy, RBP captures the probability that the correct token ranks among the top predictions, which is not only a nature performance metric but also a critical in practical scenarios such as greedy decoding or top-k sampling. Extensive experiments across multiple datasets and model families show that this law reliably characterizes performance improvement with scale. Our results demonstrate that Relative-based Scaling Law complements cross-entropy scaling laws. It provides new insights into emergence phenomena and offers an intriguing open question for theoretical studies about scaling laws.

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

## A APPENDIX

## A EXPERIMENTAL SETUP

**Models.** We select several representative model families covering a wide range of scales. Specifically, we include the `Pythia` series (Biderman et al., 2023) (14M–12B), the `GPT-2` series (Radford et al., 2019), the `OPT` family released by Meta (Zhang et al., 2022), and the more recent `Qwen2.5` series (Qwen Team et al., 2024) (0.5B–14B). Together these span over four orders of magnitude in parameter count and provide multiple independently trained scaling series.

**Datasets.** To ensure robustness across domains, we evaluate on four representative benchmarks spanning diverse linguistic and reasoning tasks: `Wikipedia` for open-domain natural text and general language modeling ability (Wikimedia Foundation), `HumanEval` for Python programming tasks focusing on code generation (OpenAI, 2021), `HotpotQA` for multi-hop question answering that requires handling complex queries (Yang et al., 2018), and the `Open Australian Legal Corpus` for long-form legal documents that test domain-specific expertise (Butler & contributors).

**Thresholds.** We estimate RBP$_k$ across a wide spectrum of threshold values $k$, spanning over five orders of magnitude, from very small thresholds up to values approaching $|\mathcal{V}|$. Each data point is obtained by averaging over at least $5 \cdot 10^5$ tokens to ensure statistical stability.

# B   CONNECTION BETWEEN CROSS-ENTROPY AND $-\log RBP_1$

## B.1   BASIC ASSUMPTION

In this section, we assume that the token probabilities $p_k$ follow a lognormal distribution:

$$p_k = \frac{\psi(k)}{c(\mu,\sigma)}, \quad \text{where} \quad \psi(x) = \frac{1}{\sqrt{2\pi}\,\sigma\, x} \exp\Big(-\frac{(\ln x - \mu)^2}{2\sigma^2}\Big), \quad x > 0, \tag{10}$$

and $c(\mu,\sigma)$ is the normalizing factor:

$$c(\mu,\sigma) = \sum_{k=1}^{\infty} \psi(k). \tag{11}$$

Define the standard normal pdf and cdf as

$$\varphi(t) = \frac{1}{\sqrt{2\pi}} e^{-t^2/2}, \qquad \Phi(t) = \int_{-\infty}^{t} \varphi(u)\, du, \tag{12}$$

and let

$$a = -\frac{\mu}{\sigma}, \qquad P = \int_{1}^{\infty} \psi(x)\, dx = \Phi\Big(\frac{\mu}{\sigma}\Big). \tag{13}$$

Conveniently, $\psi(1) = \varphi(a)/\sigma$.

We also define the truncated moments (using $t = \ln x$):

$$I_1 \equiv \int_{1}^{\infty} \psi(x)\, \ln x\, dx = \mu P + \sigma\varphi(a), \tag{14}$$

$$I_2 \equiv \int_{1}^{\infty} \psi(x)\, \ln^2 x\, dx = (\mu^2 + \sigma^2)P + \mu\sigma\varphi(a). \tag{15}$$

## B.2   CALCULATING $-\log p_1$

Since $p_1 = \psi(1)/c$, with $\psi(1) = \frac{1}{\sqrt{2\pi}\sigma} e^{-\mu^2/(2\sigma^2)}$, we have

$$-\log p_1 = \log c(\mu,\sigma) + \log(\sqrt{2\pi}\sigma) + \frac{\mu^2}{2\sigma^2}. \tag{16}$$

## B.3   CALCULATING CROSS-ENTROPY LOSS

Starting from the definition,

$$\text{CE} = -\sum_{k \geq 1} p_k \log p_k = \log(\sqrt{2\pi}\sigma) + \log c(\mu,\sigma) + \frac{1}{c(\mu,\sigma)}\left[\frac{1}{2\sigma^2}\sum_k \psi(k)(\ln k - \mu)^2 + \sum_k \psi(k)\ln k\right]. \tag{17}$$

Using the integral approximations $\sum_k \psi(k)\ln k \approx I_1$, $\sum_k \psi(k)\ln^2 k \approx I_2$, and $c(\mu,\sigma) \approx \frac{1}{2}\psi(1) + P$, we obtain

$$\text{CE} \approx \log(\sqrt{2\pi}\sigma) + \log c + \frac{1}{c}\left[\frac{1}{2\sigma^2}(I_2 - 2\mu I_1 + \mu^2 c) + I_1\right]. \tag{18}$$

Substituting $(I_1, I_2)$ gives

$$\text{CE} \approx \log(\sqrt{2\pi}\sigma) + \log c + \frac{\mu^2}{2\sigma^2} + \frac{1}{c}\left[P\Big(\frac{\mu^2 + \sigma^2}{2\sigma^2} + \mu\Big) + \varphi(a)\Big(\frac{\mu}{2\sigma} + \sigma\Big) - \frac{\mu}{\sigma^2}(\mu P + \sigma\varphi(a))\right], \tag{19}$$

with

$$c \approx \tfrac{1}{2}\psi(1) + P = \tfrac{1}{2}\frac{\varphi(a)}{\sigma} + P. \tag{20}$$

Simplifying yields

$$\mathrm{CE} \approx \log(\sqrt{2\pi}\sigma) + \log c + \frac{\mu^2}{2\sigma^2} + \frac{1}{c}\left[ P\Big(\frac{\sigma^2 - \mu^2}{2\sigma^2} + \mu\Big) + \varphi(a)\Big(\sigma - \frac{\mu}{2\sigma}\Big)\right]. \tag{21}$$

### B.4 Discussion

As $|\mu|$ increases, the term $\frac{\mu^2}{2\sigma^2}$ dominates CE, which is also true for $-\log(p_1)$. This explains why CE and $-\log RBP_1$ exhibit similar scaling behavior as the model grows.

Empirically, we observe that $\mu/\sigma$ scales approximately linearly with model size $S$, implying that both CE and $-\log \mathrm{RBP}_1$ follow a power-law with $S$ and share similar forms and exponents.

