# OpenReview forum: "Relative-Based Scaling Law for Neural Language Models"
_ICLR.cc/2026/Conference — ICLR 2026 Conference Withdrawn Submission_

### Official Review · Reviewer_QDHJ · 2025-10-29

**Soundness:** 2
**Presentation:** 3
**Contribution:** 2
**Rating:** 2
**Confidence:** 5

**Summary:**

In this paper, the authors introduce a new metric Relative-Based Probability (RBP) that captures a language model's ability to rank the correct token among top candidates. The authors try to derive scaling laws (power laws) between model parameter count and RBP. They also use the derived relationships to explain the "emergence" phenomenon.

**Strengths:**

- The new scaling law approach that quantifies relative ranking of tokens which is important for top-k sampling.
- Explanation of "emergence" using RBP.
- The authors released the code that can be used for reproduction of their findings.

**Weaknesses:**

- No held-out points validation, no confidence intervals.
- The scaling laws should be fitted not on the all data but on the Pareto frontier [1, 2] (e.g. dividing compute axis in bins taking the points with minimal loss for each bin). The points that were used for the fits in the paper are not necessarily on the Pareto frontier so at best they are not scaling law but scaling trends.
- Number of points for the fit is < 10 and for all of the fits except Pythia they are <5. The sample size is too small to make any conclusion for the power law relationship and makes all claims unsubstantial.

1. Hoffmann, Jordan, et al. "Training compute-optimal large language models." _arXiv preprint arXiv:2203.15556_ (2022).
2. Kaplan, Jared, et al. "Scaling laws for neural language models." _arXiv preprint arXiv:2001.08361_ (2020).

**Questions:**

- Why do you think Relative-based Scaling Law should be derived, considering the Section 5.2 main result "the exponents of fitted cross-entropy based power laws and RBP-based ones are almost identical"?

---

### Official Review · Reviewer_1dys · 2025-10-30

**Soundness:** 1
**Presentation:** 2
**Contribution:** 2
**Rating:** 2
**Confidence:** 4

**Summary:**

The paper studies neural language models' scaling law using a different base metric, a novel relative-based metric instead of the common cross-entropy loss. The motivation is the CE loss may not capture rank change of the ground truth token, which is important in top-k style sampling.

The findings of the paper are: (1) scaling law consistent with CE loss holds for rank-based metric as well; (2) proposed an explanation of emergent behavior using scaling law based on rank-based metric.

**Strengths:**

The paper is written well: it explains the motivation clearly, has a clear focus, and is easy to read and follow.

**Weaknesses:**

1. The main weakness is that, while the relative-based metric can measure different aspects of model performance compared to cross entropy loss, as clearly shown in Figure 1, the experiments and findings did not uncover novel knowledge of scaling behavior distinct from those know from CE loss. As such, many readers like me would interpret the paper's significance as a confirmation of the scaling law, previously known w.r.t. to the CE loss, now to the rank-based metric as well. But again the confirmation may come as expected to many, because while CE loss and rank-based metric are distinct and might be inconsistent in small changes, they are much more consistent and correlated in large changes, for example probability of correct token grows from 10% to 90%.

In summary, while the paper starts with an interesting motivation and setup, it fails to uncover novel insights of the scaling law from this setup.

2. Section 5.1 does not give a convincing explanation of emergent behavior in LLM to me. The authors construct a task success probability based on the rank-based metric, and by that construction, the task success probability must have this "emergent" or sharp transition curve in Figure 7. But the most important weakness here is how does the self-defined task success probability relate to capability being learned in LLM, which the emergent phenomenon cares about? The fact is It does not have any relation because the task success probability is simply the rank-based metric to the power of N. The assumption of "independence and stationarity across positions" in line 370 clearly does not hold in LLM generations as well.

Smaller questions:
1. I feel the hyper parameter of k in rank based metric makes it a less suitable metric to consider in scaling laws because it introduces an additional hyper parameters and unnecessarily complicates the study of the fundamental scaling law.
2. In sections 4.1 and 4.2, the authors refers to both Figure 5, but they should be Figure 3 and 4 instead.

**Questions:**

See above.

---

### Official Review · Reviewer_aguA · 2025-11-01

**Soundness:** 3
**Presentation:** 3
**Contribution:** 2
**Rating:** 4
**Confidence:** 3

**Summary:**

This paper proposes a new scaling law based on relative-based probability (RBP), which evaluates models by the relative order of correct answers by measuring whether the correct answer is in the top k predicted answers for some k. This scaling law and metric are in contrast with those based on cross-entropy. Experiments for different values of k show that a precise power-law law based on RBP can explain performance improvements for lower values of k, but that the trend break down for larger values of k. They also argue that emergence can be explained through the lens of RBP.

**Strengths:**

- The paper is well-written, and takeaways are easy to understand.
- The proposed RBP metric addresses a key limitation of cross-entropy as a metric and better aligns with real-world inference practices, which often involve greedy decoding or top k sampling.

**Weaknesses:**

- Previous works have put forth the view that whether model performance exhibits "emergence" depends on the metric being measured (Schaeffer et al., 2023); it is thus unclear what new information has been learned by proposing a specific metric that can explain away emergence.
- The paper would be strengthened by showing that the RBP based scaling law leads to new insights about how model performance scales (for example, are there different compute-optimal data-to-parameter ratios?) that would not be explained by cross-entropy based laws, but it is not clear what such insights are.

**Questions:**

- Authors write in the abstract that RBP and the associated scaling law offer insights for practical model development; what are some of these practical insights?

Minor Notes:
- Line 242: Should say Figure 3 and not Figure 5
- Line 312: "investigated" -> "investigated."

---

### Official Review · Reviewer_RXhj · 2025-11-09

**Soundness:** 3
**Presentation:** 3
**Contribution:** 2
**Rating:** 4
**Confidence:** 3

**Summary:**

The paper proposes a new scaling methodology that focuses on the power-law relationship between the relative ranking of tokens and the model's parameter size. This approach offers an interpretation for the emergence phenomenon.

**Strengths:**

Its strength lies in the novelty of the proposed idea and its substantiation by a large body of experimental results.

**Weaknesses:**

Objectively, the paper’s contribution is limited. Although it provides a new scaling perspective, the conclusions derived do not appear to be significantly more informative or useful compared to those from the traditional log scaling law.

**Questions:**

As listed in weakness

---

### Note · Authors · 2025-11-22

I have read and agree with the venue's withdrawal policy on behalf of myself and my co-authors.